# Development of an artificial intelligence-based algorithm to classify images acquired with an intraoral scanner of individual molar teeth into three categories

**Nozomi Eto**[1,2]*, **Junichi Yamazoe**[3], **Akiko Tsuji**[2], **Naohisa Wada**[1,4], **Noriaki Ikeda**[2]

**1** Division of Interdisciplinary Dentistry, Graduate School of Dental Science, Kyushu University, Fukuoka, Japan, **2** Department of Forensic Pathology and Sciences, Graduate School of Medical Sciences, Kyushu University, Fukuoka, Japan, **3** Section of Geriatric Dentistry and Perioperative Medicine in Dentistry, Kyushu University Hospital, Fukuoka, Japan, **4** Division of General Dentistry, Kyushu University Hospital, Fukuoka, Japan

* eto@dent.kyushu-u.ac.jp

**Data Availability Statement:** All relevant data are within the paper.

**Funding:** The authors received no specific funding for this work.

## Abstract

### Background

Forensic dentistry identifies deceased individuals by comparing postmortem dental charts, oral-cavity pictures and dental X-ray images with antemortem records. However, conventional forensic dentistry methods are time-consuming and thus unable to rapidly identify large numbers of victims following a large-scale disaster.

### Objective

Our goal is to automate the dental filing process by using intraoral scanner images. In this study, we generated and evaluated an artificial intelligence-based algorithm that classified images of individual molar teeth into three categories: (1) full metallic crown (FMC); (2) partial metallic restoration (In); or (3) sound tooth, carious tooth or non-metallic restoration (CNMR).

### Methods

A pre-trained model was created using oral-cavity pictures from patients. Then, the algorithm was generated through transfer learning and training with images acquired from cadavers by intraoral scanning. Cross-validation was performed to reduce bias. The ability of the model to classify molar teeth into the three categories (FMC, In or CNMR) was evaluated using four criteria: precision, recall, F-measure and overall accuracy.

### Results

The average value (variance) was 0.952 (0.000140) for recall, 0.957 (0.0000614) for precision, 0.952 (0.000145) for F-measure, and 0.952 (0.000142) for overall accuracy when the algorithm was used to classify images of molar teeth acquired from cadavers by intraoral scanning.

**Competing interests:** The authors have declared that no competing interests exist.

## Conclusion

We have created an artificial intelligence-based algorithm that analyzes images acquired with an intraoral scanner and classifies molar teeth into one of three types (FMC, In or CNMR) based on the presence/absence of metallic restorations. Furthermore, the accuracy of the algorithm reached about 95%. This algorithm was constructed as a first step toward the development of an automated system that generates dental charts from images acquired by an intraoral scanner. The availability of such a system would greatly increase the efficiency of personal identification in the event of a major disaster.

## 1 Introduction

Dental evidence is widely used for personal identification because the teeth exhibit age-related changes, have features that are unique to an individual, and resist decomposition after death [1, 2]. Forensic dentistry identifies deceased individuals by collecting postmortem data, such as dental charts, oral-cavity pictures and dental X-ray images, and comparing them with ante-mortem records [3]. A notable disadvantage of this approach is that it is a slow process due to the time needed to perform the postmortem examinations, manually interpret the findings and compare them with antemortem records. Hence, the use of conventional techniques can lead to delays in victim identification following major disasters such as earthquakes that claim many lives. The development of an automated system that produces dental charts from digitized images of the teeth would greatly speed up the process of victim identification after a large-scale disaster.

Previous studies aimed at improving personal identification methods have evaluated superposition of reconstructed images of the palatal rugae [4], superposition of computed tomography (CT) images of the skull [5, 6], and superposition of dental X-ray images [7]. However, obtaining X-ray or CT images from a huge number of victims is not only time-consuming but also difficult to achieve at the site of a disaster because it requires radiation-generating equipment that is not easily portable. Although oral-cavity pictures potentially could be used as part of an automated system for victim identification, it is often challenging to obtain postmortem pictures with a dental camera due to the presence of rigor mortis that restricts mouth opening. By contrast, real-time imaging of the teeth can be performed easily and rapidly with an intraoral scanner even when there is some restriction of mouth opening. Furthermore, an intraoral scanner is a handheld device that is highly portable, making it well suited for use after a large-scale disaster.

The development of a system that automatically creates dental charts from images acquired with an intraoral scanner would greatly facilitate the rapid creation of postmortem dental charts for personal identification in the event of a major disaster. In this study, as a component of automated dental chart filling system, we generated and evaluated an artificial intelligence (AI)-based algorithm that analyzes images of molar teeth acquired with an intraoral scanner and classifies each tooth as one of three types: (1) full metallic crown (FMC); (2) partial metallic restoration (In); or (3) sound tooth, carious tooth or non-metallic restoration (CNMR).

## 2 Materials and methods

### 2.1 Ethics

In this study, existing information was anonymized and used unidentified. The anonymized correspondence table was stored in a separate file in a separate location. We do not give

individual written or oral informed consent because we are using existing information. Therefore, we have disclosed the information of this research on our website and provided contact information so that patients or their families can decline if they do not wish to be eligible. This study was approved by the Kyushu University Certified Institutional Review Board for Clinical Trials (reference no. 2020–499). All experiments were conducted in accordance with approved guidelines.

## 2.2 Intraoral scanning

Images were obtained using a TRIOS third-generation intraoral scanner (3Shape, Copenhagen, Denmark). The TRIOS scanner does not require opacification (powder-free) and relies on the principle of confocal microscopy to acquire a sequence of video images using structured illumination from a light-emitting diode. The imaging system produces a colored visualization of the scanned structures. The scanner's acquisition software generates proprietary files (DICOM, Digital Imaging and Communications in Medicine format) that can be exported to an open file format (STL, Standard Tessellation Language).

Since the overarching objective of our research is to develop a system for use in disaster victim identification, this study was conducted on cadavers. The oral-cavities of 34 cadavers to be dissected were imaged with an intraoral scanner at the Department of Forensic Pathology and Sciences, Graduate School of Medical Sciences, Kyushu University, Fukuoka, Japan. First, the oral-cavity of the cadaver was cleaned to remove any contaminants (including body fluids) that might hinder the collection of accurate findings. Then, the head of the scanner was inserted into the oral-cavity between the upper and lower teeth and slowly moved along the teeth to acquire the images. The minimal opening required to allow the scanner head to be inserted into the oral-cavity was around 2 cm (the thickness of the scanner head). The three-dimensional images of the upper and lower teeth were saved as STL files, and occlusal views of the teeth were generated with a snipping tool and saved as Portable Network Graphic (PNG) files. ImageJ (National Institutes of Health, Bethesda, MD, USA) was used to generate individual images of the upper and lower molars (256×256 pixels) from the occlusal views of the teeth (examples are shown in Fig 1).

## 2.3 Classification of the extracted images

The individual molar teeth extracted from the scanned images were classified as FMC (full metallic crown), In (partial metallic restoration) or CNMR (sound tooth, carious tooth or non-metallic restoration). Images that were unclear on visual inspection were excluded from the analysis.

## 2.4 Deep learning machine

The deep learning machine used in this study was comprised of a Core i9-7920X central processing unit (2.90 GHz, 12 cores, 24 threads; Intel, Santa Clara, CA, USA), TITAN V graphics processing unit (32 GB, 5120 cores, 4096-bit bus width; Nvidia, Santa Clara, CA, USA) and Ubuntu 16.04.5 LTS software (Canonical, London, UK).

## 2.5 Network architecture

The convolutional neural network (CNN) was based on the LeNet architecture proposed by LeCun et al. [8]. The LeNet architecture consisted of two convolution layers, two max-pooling layers and two affine layers (Fig 2). When color molar images were provided as the input to the CNN, the output of the network classified the tooth condition as one of three types (FMC,

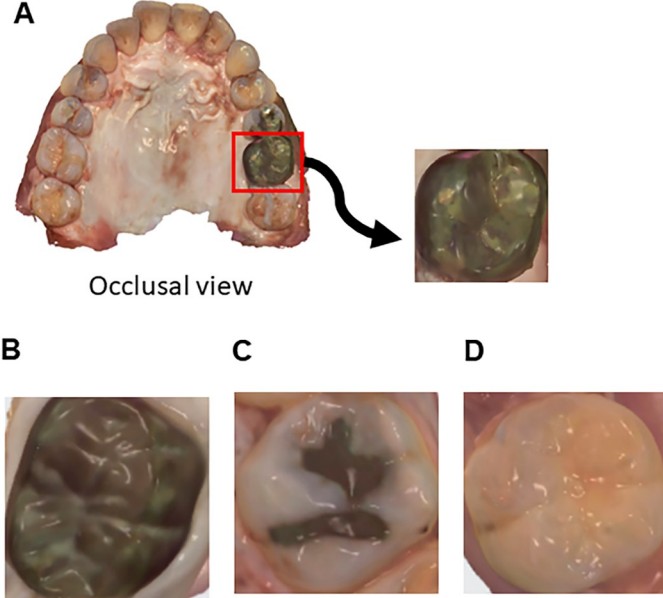

**Fig 1. Representative images of individual upper and lower molar teeth (occlusal views) extracted from images obtained with the intraoral scanner.** A. Occlusal view generated by intraoral scanning. Images of individual molar teeth were extracted from the occlusal view. B. full metallic crown (FMC). C. partial metallic restoration (In). D. sound tooth, carious tooth or non-metallic restoration (CNMR).

In or CNMR). The following hyperparameters were used in this study: training epochs: 30; base learning rate: 0.001; and training / validation rate: 80% / 20%.

## 2.6 Generation of a pre-trained model

Deep learning in general is considered to require a large number of training samples [9]. Since there was a limitation to the number of cadavers available for this research, the present study utilized the transfer learning technique, which is a method of transferring knowledge across domains to overcome problems associated with small training datasets [10]. Therefore, we carried out a pre-training process using oral-cavity pictures obtained from patients who had attended the dentistry section of Kyushu University Hospital. Training data (i.e., occlusal views of individual molar teeth) were extracted from the oral-cavity pictures using the method described in section 2.2 (Intraoral scanning) (Fig 3), and each tooth was classified as FMC, In or CNMR as described above. Images that were unclear on visual inspection were excluded. 300 images of each category were randomly divided into 10 folders of 30 images each, and training and test runs of the CNN were performed a total of 10 times for cross-validation.

## 2.7 Generation of a classification model

The classification model was generated by applying transfer learning to the pre-trained model using images obtained from the cadavers by the intraoral scanner. 60 images of each tooth category (FMC, In or CNMR) were randomly divided into 4 folders of 15 images, and a total of 4 training and test runs of the CNN were performed for cross-validation.

## 2.8 Two-step cascade method for classification of tooth condition

An additional trained model was generated using a two-step cascade method to classify tooth condition. Images obtained from cadavers by intraoral scanner were used. In the first step, the

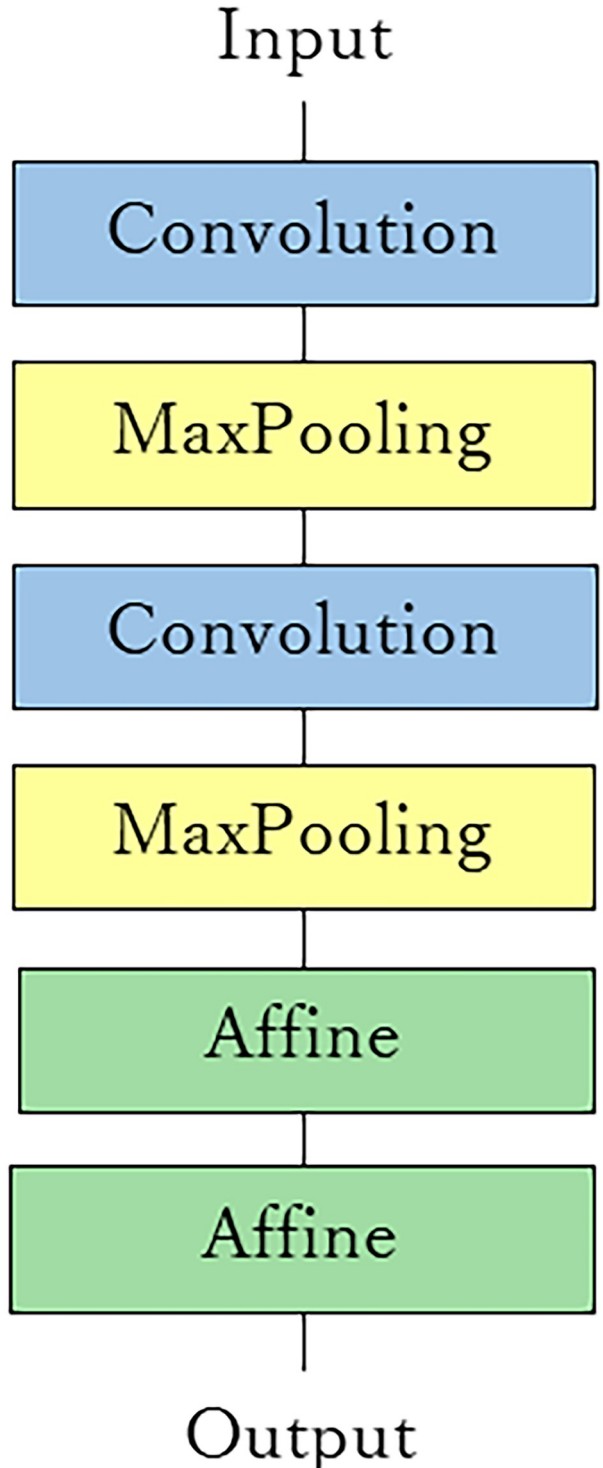

**Fig 2. Architecture of the LeNet used in our study.** The LeNet architecture consisted of two convolution layers, two max-pooling layers and two affine layers.

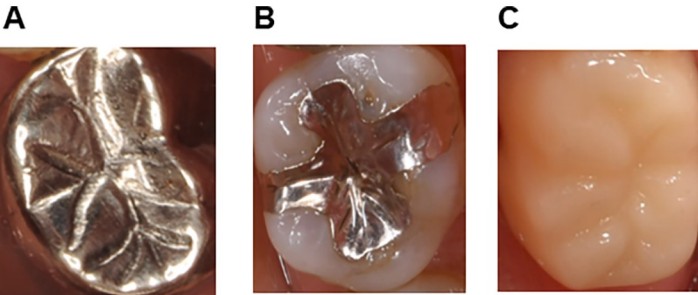

**Fig 3. Representative images of individual upper and lower molar teeth (occlusal views) extracted from oral-cavity pictures.** A. full metallic crown (FMC). B. partial metallic restoration (In). C. sound tooth, carious tooth or non-metallic restoration (CNMR).

tooth was classified as 'presence or absence of metallic restoration'. In the second step, 'metallic restoration' was sub-classified as 'full or partial' (Fig 4).

## 2.9 Evaluation of the models

The ability of each model to classify tooth condition was evaluated using four criteria: precision, recall, F-measure and overall accuracy. Precision was calculated as the fraction of correct predictions for a certain class. Recall was calculated as the fraction of instances of a class that were correctly predicted. F-measure was defined as the harmonic mean of precision and recall: F-measure = (2 × precision × recall) / (recall + precision). Overall accuracy was calculated as the fraction of instances that were correctly classified.

## 3 Results

### 3.1 Evaluation of the pre-trained model

Table 1 shows the recall, precision, F-measure and overall accuracy values for each of 10 cross-validated tests using oral-cavity pictures obtained from patients. The average value (variance) was 0.937 (0.00128) for recall, 0.941 (0.00108) for precision, 0.937 (0.00129) for F-measure, and 0.939 (0.00107) for overall accuracy (values given to 3 significant figures).

### 3.2 Evaluation of the classification model

The classification algorithm was generated through transfer learning using images obtained from cadavers by intraoral scanning. Table 2 shows the recall, precision, F-measure and overall

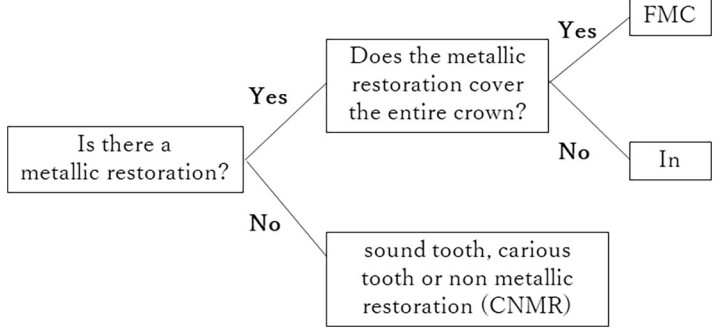

**Fig 4. Schematic diagram of the two-step cascade classification.** FMC: full metallic crown; In: partial metallic restoration; CNMR: sound tooth, carious tooth or non-metallic restoration.

**Table 1. Recall, precision, F-measure and overall accuracy for each test at the pre-trained model.**

| Test | Recall | Precision | F-measure | Accuracy |
|---|---|---|---|---|
| 1 | 0.8973 | 0.9091 | 0.8970 | 0.8988 |
| 2 | 0.8466 | 0.8579 | 0.8462 | 0.8588 |
| 3 | 0.9659 | 0.9662 | 0.9658 | 0.9662 |
| 4 | 0.9544 | 0.9579 | 0.9548 | 0.9550 |
| 5 | 0.9436 | 0.9442 | 0.9432 | 0.9438 |
| 6 | 0.9655 | 0.9696 | 0.9659 | 0.9662 |
| 7 | 0.9321 | 0.9317 | 0.9317 | 0.9325 |
| 8 | 0.9555 | 0.9607 | 0.9553 | 0.9555 |
| 9 | 0.9555 | 0.9607 | 0.9553 | 0.9555 |
| 10 | 0.9555 | 0.9561 | 0.9551 | 0.9555 |
| Average | 0.9372 | 0.9414 | 0.9370 | 0.9388 |
| Variance | $1.28 \times 10^{-3}$ | $1.08 \times 10^{-3}$ | $1.29 \times 10^{-3}$ | $1.07 \times 10^{-3}$ |

**Table 2. Recall, precision, F-measure, and overall accuracy for each test at the classification model.**

| Test | Recall | Precision | F-measure | Accuracy |
|---|---|---|---|---|
| 1 | 0.9333 | 0.9444 | 0.9326 | 0.9333 |
| 2 | 0.9555 | 0.9583 | 0.9539 | 0.9545 |
| 3 | 0.9659 | 0.9662 | 0.9658 | 0.9662 |
| 4 | 0.9544 | 0.9579 | 0.9548 | 0.9550 |
| Average | 0.9523 | 0.9567 | 0.9518 | 0.9523 |
| Variance | $1.40 \times 10^{-4}$ | $6.10 \times 10^{-5}$ | $1.44 \times 10^{-4}$ | $1.41 \times 10^{-4}$ |

accuracy values for each of 4 cross-validated tests. The average value (variance) was 0.952 (0.000140) for recall, 0.957 (0.0000614) for precision, 0.952 (0.000145) for F-measure, and 0.952 (0.000142) for overall accuracy.

### 3.3 Evaluation of the two-step cascade classification model

Table 3 (step 1 of the two-step cascade classification model) and Table 4 (step 2 of the two-step cascade classification model) present the recall, precision, F-measure and overall accuracy values for each of 4 cross-validated tests using images acquired from cadavers with the intraoral scanner. Step 1 of the two-step cascade classification model (i.e., presence or absence of a metallic restoration) achieved particularly high values (greater than 0.987) for recall, precision, F-measure and overall accuracy.

**Table 3. Recall, precision, F-measure, and overall accuracy for each test using the first step of the two-step model.**

| Test | Recall | Precision | F-measure | Accuracy |
|---|---|---|---|---|
| 1 | 1.000 | 1.000 | 1.000 | 1.000 |
| 2 | 0.9827 | 0.9687 | 0.9750 | 0.9772 |
| 3 | 1.000 | 1.000 | 1.000 | 1.000 |
| 4 | 0.9666 | 0.9838 | 0.9745 | 0.9777 |
| Average | 0.9873 | 0.9881 | 0.9874 | 0.9887 |
| Variance | $1.93 \times 10^{-4}$ | $1.70 \times 10^{-4}$ | $1.60 \times 10^{-4}$ | $1.27 \times 10^{-4}$ |

**Table 4. Recall, precision, F-measure, and overall accuracy for each test using the second step of the two-step model.**

| Test | Recall | Precision | F-measure | Accuracy |
|---|---|---|---|---|
| 1 | 0.9000 | 0.9166 | 0.8989 | 0.9000 |
| 2 | 0.9666 | 0.9666 | 0.9654 | 0.9655 |
| 3 | 0.9333 | 0.9411 | 0.9330 | 0.9333 |
| 4 | 0.9666 | 0.9687 | 0.9666 | 0.9666 |
| Average | 0.9416 | 0.9483 | 0.9410 | 0.9414 |
| Variance | $7.60 \times 10^{-4}$ | $4.50 \times 10^{-4}$ | $7.70 \times 10^{-4}$ | $7.40 \times 10^{-4}$ |

## 4 Discussion

Dental evidence has been used for personal identification during recent natural disasters. However, many dentists have little or no experience of collecting dental information from cadavers and experience psychological distress when faced with this task [9]. The above factors can lead to errors in judgment as well as limitations in manpower availability for personal identification of victims after a large-scale disaster. Although previous reports have described the automatic interpretation of dental findings and the collation of ante/postmortem images [11, 12], these studies were based on X-rays or CT scans of the cadavers, which are time-consuming to perform and require specialized equipment that generates radiation and lacks portability. As a method that does not use them, in the present study, images of individual molar teeth were acquired with an intraoral scanner, and an algorithm was created to categorize the tooth in each image as FMC, In or CNMR. Notably, the algorithm was able to classify molar teeth with an accuracy of around 95%.

The intraoral scanner utilized in this study has several advantageous features that make it well suited to the acquisition of dental evidence from cadavers. First, the captured images are displayed in real time on a dedicated application. Second, the images can be saved as digital data for later analysis. Third, the scanner used in this study comes with RealColor™ Technology that allows it to accurately reproduce color tones [13]. Fourth, the TRIOS scanner does not require opacification (powder-free) because it is equipped with technology that controls for the reflection of light by metals and other substances; hence, this scanner is easier to use than early-generation scanners that require the application of powder for opacification. Finally, the intraoral scanner is a highly portable, handheld device with a small scanning head that can be inserted through an opening of only 2 cm. Thus, unlike taking oral-cavity pictures with a dental camera, intraoral scanning can be performed even when mouth opening is somewhat restricted by rigor mortis or other factors. Based on the above features, we believe that the use of an intraoral scanner would improve the accuracy and speed of dental evaluation in the event of a large-scale disaster.

In this study, the number of images we were able to obtain by intraoral scanning of cadavers was limited. Possible methods to overcome this limitation include data augmentation, increasing the size of the training sample [9] and transfer learning [14]. We performed transfer learning in this study because of the availability of a large number of oral-cavity pictures that were similar to the images acquired with the intraoral scanner. First, the CNN was pre-trained using oral-cavity pictures, which were relatively easy to collect. Then, using the parameter of pre-trained model as an initial parameter of classification model, we generated the classification model using images obtained by intraoral scanning of cadavers. This approach allowed us to construct a highly accurate CNN using only a small number of intraoral scans.

The use of dental evidence for the personal identification of disaster victims requires that the dental findings are collected accurately [15]. Since teeth with composite resin restorations

can sometimes be mistaken for non-restored teeth even when observed with the naked eye, the present study limited the classification of molar tooth condition to only three categories (i.e., FMC, In and CNMR) that exhibit obvious differences in color. The accuracy of the algorithm was around 95% when these three categories were used and reached about 98% when the classification was based only on the presence or absence of a metal-colored restoration (Table 3), which suggests that the accuracy of the algorithm can be improved by simplifying the conditions. However, in order to be utilized for personal identification purposes, the algorithm will need to be improved so that it can recognize non-metallic restorations and thereby provide a more complex classification. We envisage that additional research will allow the algorithm to be further developed so that it not only recognizes a wider range of tooth features (including both metallic and non-metallic restorations) but also identifies individual teeth in an occlusal view. The creation of such an algorithm would allow dental charts to be automatically generated from occlusal views obtained by intraoral scanning.

In conclusion, we have developed an AI-based algorithm that can analyze images acquired with an intraoral scanner and classify molar teeth into one of three types (FMC, In or CNMR) based on the presence/absence of metallic restorations. Furthermore, the accuracy of the algorithm reached about 95%. This algorithm was created as a first step toward the construction of a system that can automatically generate a dental chart from images obtained with an intraoral scanner. The development of such an automated system would greatly improve the efficiency of personal identification in the event of a major disaster.

## Acknowledgments

The authors would like to thank Dr Kenichi Morooka, Professor, Graduate School of Natural Science and Technology, Okayama University for helpful discussion and comments on the manuscript. We are also grateful to Dr Yongsu Yoon, Assistant Professor, Department of Radiological Science, Dongseo University for help with the image editing and deep learning methods. We thank OXMEDCOMMS (www.oxmedcomms.com) for writing assistance.

## Author Contributions

**Conceptualization:** Junichi Yamazoe.

**Formal analysis:** Nozomi Eto.

**Funding acquisition:** Noriaki Ikeda.

**Investigation:** Nozomi Eto.

**Methodology:** Nozomi Eto.

**Project administration:** Junichi Yamazoe.

**Resources:** Noriaki Ikeda.

**Supervision:** Junichi Yamazoe, Noriaki Ikeda.

**Validation:** Junichi Yamazoe.

**Writing – original draft:** Nozomi Eto.

**Writing – review & editing:** Junichi Yamazoe, Akiko Tsuji, Naohisa Wada, Noriaki Ikeda.

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
