## [Decision Letter · Decision Letter 0]

8 Nov 2021

PONE-D-21-30844Development of an artificial intelligence-based algorithm to evaluate tooth condition from images acquired with an intraoral three-dimensional scannerPLOS ONE

Dear Dr. ETO,

Thank you for submitting your manuscript to PLOS ONE. After careful consideration, we feel that your work has merit but does not fully meet the journal’s publication criteria in the form as it currently stands. Therefore, we invite you to submit a revised version of the manuscript that addresses the points raised during the review process. Your manuscript has been evaluated by two experts in the field and their comments are attached here below for your reference. During the revision some significant concerns have emerged, and in particular:The main outcomes of the work are not clearly exposed;Further details are needed in different sections of the manuscript: both reviewers agree on this point, please check their comments carefully;Abstract and Introduction need thorough restructuring;The limitations of this research should be described more in depth. Please submit your revised manuscript by Dec 23 2021 11:59PM. If you will need more time than this to complete your revisions, please reply to this message or contact the journal office at plosone@plos.org. Please include the following items when submitting your revised manuscript:A rebuttal letter that responds to each point raised by the academic editor and reviewer(s). You should upload this letter as a separate file labeled 'Response to Reviewers'.A marked-up copy of your manuscript that highlights changes made to the original version. You should upload this as a separate file labeled 'Revised Manuscript with Track Changes'.An unmarked version of your revised paper without tracked changes. You should upload this as a separate file labeled 'Manuscript'.

We look forward to receiving your revised manuscript.

Kind regards,

Francesco Bianconi, Ph.D.

Academic Editor

PLOS ONE

Journal Requirements:

a) Did participants provide their written or verbal informed consent to participate in this study?

Reviewers' comments:

Reviewer's Responses to Questions

**Comments to the Author**

1. Is the manuscript technically sound, and do the data support the conclusions?

Reviewer #1: Yes

Reviewer #2: Yes

2. Has the statistical analysis been performed appropriately and rigorously? 

Reviewer #1: Yes

Reviewer #2: Yes

3. Have the authors made all data underlying the findings in their manuscript fully available?

Reviewer #1: Yes

Reviewer #2: Yes

4. Is the manuscript presented in an intelligible fashion and written in standard English?

Reviewer #1: Yes

Reviewer #2: Yes

5. Review Comments to the Author

Reviewer #1: In this work an AI-based algorithm has been proposed for the automatic identification of the condition of molar teeth from cadavers. The main aim of this activity was to provide a first step for the construction of a digitized dental chart from imaging data, in support of post-mortem identification analyses in forensic dentistry.

The paper is well written and has a good general organisation among sections. The main issue that the reviewer can see, is that the research seems to be not concluded as it is presented, and the main outcomes of this activity are not clearly exposed. The authors stated that this is the first part of a wider project and research, but the novelty of the proposed methodology and how it is placed in relation to the state of the art is not defined. Moreover, it should be better presented throughout the work how this part of the research activity is or will be linked to the whole project.

The reviewer suggests major revisions in order to improve the article impact.

INTRODUCTION

The Introduction section should be improved better clarifying the potentiality and the novelty of the proposed approach, in order to classify it in relation to the most used techniques.

MATERIALS AND METHODS

The ‘Intraoral 3D scanning’ section describes mainly the characteristics of the scanner, but there is no information on the experimental setup and procedure followed for the data acquisition from cadavers. In the Discussion section it is stated that the intraoral 3D scanning can be used also in challenging situations, typical when the mouth opening width is limited because of rigor mortis. For this reason, in the ‘Materials and Methods’ section, a step-by-step description of the procedure used for the 3D scanning acquisitions on cadavers should be reported, also highlighting, if there have been, difficulties faced.

Line 89: dcm is the file extension. The file format is called DICOM.

Line 93: how many cadavers were available for the analysis? The only information seems to be that sixty images were available for the classification model (line 142).

Figure 1: this figure shows an example of the three molar teeth types, but for the model implementation it is stated that you have used scans. So, the reviewer is wondering if these scans were exported in some CAD format (usually stl) and then these used for the analysis. This point is not clear. For this reason, the reviewer suggests to show some images representing DICOM or stl models (or better both of them), in order to clarify the type of available data. Moreover, if some scans post-processing was performed this should be reported in the Methods section as well as the accuracy of the model (number of elements, texture, and so on).

Lines 126-127: here the authors stated that the number of available scanning from cadavers was not sufficient for the training of the network. Is it possible to sustain this with some bibliography and provide information on the proper number of training data?

Figure 4: this image is not very useful; indeed, the two-step cascade process is clear as it is explained in this section. It would be more beneficial for the reader if a comprehensive workflow of the overall process and AI-based algorithm is provided with all the main steps clearly described.

RESULTS

Lines 167-168: this sentence in not comprehensible; please rephrase.

DISCUSSION

Lines 201-202: It is not clear which features have been used for the identification of these conditions: texture, shape, volume? In the Methods section a more detailed description of the available features should be provided along with the features used within the AI architecture.

Lines 205-206: How the model can be used for victim recognition based only on the recognition of the molar teeth condition? This point is not clear to the reviewer. The authors have described the full process for the creation and training of the network but its application for the specific case of victim recognition remains vague.

Lines 233-234: the possible use of morphological characteristics for victim recognition is undoubtedly an improvement for the model, but how do you link the work here presented with the possibility to use new features for victim recognition? In general, the connection between the part of the work here presented and the overall project has to be better clarified and carried on throughout the paper.

Reviewer #2: The paper is interesting and worth publishing.

The abstract should be reworded and more clearly structured.

For the readers unfamiliar with intraoral scanners more information should be provided on the applied scanner and explanation why it is powder free should be added.

The main limitation of the study is a relatively low number of cases, especially from the intraoral scanner. However, preliminary results are promising.

Another limitation of the study is that the authors teach the algorithm to distinguish between 3 groups: 1. full metallic crown 2. partial metallic restoration and 3. sound tooth, caries tooth and tooth with non-metallic restoration merged in one group. Probably metallic restorations are prevalent in the studied Japanese population, but this is a major limitation in populations in which more esthetic restorations are applied. The authors should discuss this limitation more indepth and also provide an attempt on solution of the problem of differentiation between healthy teeth and restored teeth using composite materials mimicking sound tissues.

What did the authors mean by "treatment scars"?

The conclusions in the present form are rather an extended summary of the results, therefore should be rephrased so that they are related to the aim of the study.

6. PLOS authors have the option to publish the peer review history of their article (what does this mean?). If published, this will include your full peer review and any attached files.

Reviewer #1: No

Reviewer #2: No

---

## [Author Response · Author response to Decision Letter 0]

24 Nov 2021

We thank the reviewers for their careful reading of our manuscript and useful comments. Our responses to the reviewers’ comments are presented below. All revisions to the manuscript are highlighted in red-colored font and referred to by line number in the responses below. Please note that the changes made do not majorly affect the content, conclusions or framework of the paper.

REVIEWER #1

INTRODUCTION

Comment

The Introduction section should be improved better clarifying the potentiality and the novelty of the proposed approach, in order to classify it in relation to the most used techniques.

Response

Thank you for this excellent recommendation. We have completely rewritten the Introduction section to streamline the text and focus on the potentiality and novelty of our study in relation to currently used approaches. We refer you to the new Introduction section in the revised manuscript (lines 48–76).

MATERIALS AND METHODS

Comment

The ‘Intraoral 3D scanning’ section describes mainly the characteristics of the scanner, but there is no information on the experimental setup and procedure followed for the data acquisition from cadavers. In the Discussion section it is stated that the intraoral 3D scanning can be used also in challenging situations, typical when the mouth opening width is limited because of rigor mortis. For this reason, in the ‘Materials and Methods’ section, a step-by-step description of the procedure used for the 3D scanning acquisitions on cadavers should be reported, also highlighting, if there have been, difficulties faced.

Response

Thank you for this helpful suggestion. We have modified the Intraoral Scanning subsection of the Materials and Methods section to include more information about the technique used for intraoral scanning of cadavers (lines 96–105). We did not encounter any specific difficulties while performing intraoral scanning, hence none are described in the manuscript.

Comment

Line 89: dcm is the file extension. The file format is called DICOM.

Response

Thank you for pointing out this inadvertent error. We have revised “DCM” to “DICOM” (line 90).

Comment

Line 93: how many cadavers were available for the analysis? 

Response

A total of 34 cadavers were available for our analysis. We have added this information to the Intraoral Scanning subsection of the Materials and Methods section (line 94).

Comment

Figure 1: this figure shows an example of the three molar teeth types, but for the model implementation it is stated that you have used scans. So, the reviewer is wondering if these scans were exported in some CAD format (usually stl) and then these used for the analysis. This point is not clear. For this reason, the reviewer suggests to show some images representing DICOM or stl models (or better both of them), in order to clarify the type of available data. Moreover, if some scans post-processing was performed this should be reported in the Methods section as well as the accuracy of the model (number of elements, texture, and so on).

Response

Thank you for these important queries. The images shown in Figure 1 were obtained as follows. First, the 3D images obtained from cadavers by intraoral scanning were saved as STL files, and occlusal views of the teeth were generated with a snipping tool and saved as PNG files. Then, ImageJ was used to generate individual images of the upper and lower molars (256×256 pixels) from the occlusal views of the teeth; examples of these images are presented in Figure 1. We have revised the Intraoral Scanning subsection of the Materials and Methods section to include the above information (lines 101–105). In addition, Figure 1 has been modified to include a panel showing an occlusal view generated by intraoral scanning, from which images of individual molar teeth were extracted. The title and legend for Figure 1 have been updated accordingly (lines 107–111).

Comment

Lines 126-127: here the authors stated that the number of available scanning from cadavers was not sufficient for the training of the network. Is it possible to sustain this with some bibliography and provide information on the proper number of training data?

Response

The number of images required to train a network varies depending on the structure of the network and the complexity of the problem. Currently, no methods are available to predict the number of images needed, and there are no relevant publications that we can cite. Since only a limited number of cadavers were available to us, we made the decision to use transfer learning to overcome any potential problems that might be associated with the use of a small training dataset. We have added some information about the transfer learning technique and a supporting reference citation to the Generation Of A Pre-trained Model subsection of the Materials and Methods section (lines 136–139).

Comment

Figure 4: this image is not very useful; indeed, the two-step cascade process is clear as it is explained in this section. It would be more beneficial for the reader if a comprehensive workflow of the overall process and AI-based algorithm is provided with all the main steps clearly described.

Response

Thank you for this helpful suggestion. We have modified Figure 4 to make it easier to understand.

Comment

Lines 167-168: this sentence in not comprehensible; please rephrase.

Response

We are sorry that the meaning of the original text was unclear. We have modified this sentence (lines 178–179).

DISCUSSION

Comment

Lines 201-202: It is not clear which features have been used for the identification of these conditions: texture, shape, volume? In the Methods section a more detailed description of the available features should be provided along with the features used within the AI architecture.

Response

We are sorry that the point was unclear. We assume that this was based on the color of the tooth in the image, because a metallic restoration appears darker in the image. However, we did not use a color threshold to distinguish dark metallic regions from light non-metallic regions because good accuracy was detected without setting. Therefore, there is no additional information at this stage.

Comment

Lines 205-206: How the model can be used for victim recognition based only on the recognition of the molar teeth condition? This point is not clear to the reviewer. The authors have described the full process for the creation and training of the network but its application for the specific case of victim recognition remains vague.

Response

Thank you for this important question. The long-term aim of this research is to develop an AI-based algorithm that can automatically generate a dental chart from images acquired with an intraoral scanner. These postmortem dental charts could be compared with antemortem records to facilitate the identification of disaster victims. As a first step toward this aim, the present study has developed an algorithm to classify the condition of the molar teeth into three types. We have completely rewritten the Introduction section (see lines 49–76) and Discussion section (lines 211–264) to better explain the objective, significance and future potential of our research.

Comment

Lines 233-234: the possible use of morphological characteristics for victim recognition is undoubtedly an improvement for the model, but how do you link the work here presented with the possibility to use new features for victim recognition? In general, the connection between the part of the work here presented and the overall project has to be better clarified and carried on throughout the paper.

Response

Since the focus of the present study was the detection of dental restorations (specifically, metallic restorations) rather than morphological characteristics, we have deleted the text referring to morphological characteristics in order to avoid confusion. Furthermore, we have rewritten the Discussion section to explain more clearly that the present research is a first step toward developing an automated system for the construction of postmortem dental charts from imaging data obtained by intraoral scanning. 

REVIEWER #2

Comment

The abstract should be reworded and more clearly structured.

Response

Thank you for this helpful recommendation. We have rewritten the Abstract section of the manuscript to improve its structure (lines 20–46).

Comment

For the readers unfamiliar with intraoral scanners more information should be provided on the applied scanner and explanation why it is powder free should be added.

Response

Thank you for this useful suggestion. We have added more information about the intraoral scanner to the Intraoral Scanning subsection of the Materials and Methods section (lines 96–105) and the Discussion section (lines 222–234).

Comment

The main limitation of the study is a relatively low number of cases, especially from the intraoral scanner. However, preliminary results are promising.

Response

Since there was a limitation to the number of cadavers available for this research, we utilized the technique of transfer learning, which focuses on transferring the knowledge across domains to overcome the problems of small training datasets. Using this approach, we were able to develop our algorithm successfully despite the limited number of cadavers. We have added more details about the use of the transfer learning to the Generation Of A Pre-trained Model subsection of the Materials and Methods section (lines 1376–146) and to the Discussion section of the manuscript (lines 235¬–243).

Comment

Another limitation of the study is that the authors teach the algorithm to distinguish between 3 groups: 1. full metallic crown 2. partial metallic restoration and 3. sound tooth, caries tooth and tooth with non-metallic restoration merged in one group. Probably metallic restorations are prevalent in the studied Japanese population, but this is a major limitation in populations in which more esthetic restorations are applied. The authors should discuss this limitation more in depth and also provide an attempt on solution of the problem of differentiation between healthy teeth and restored teeth using composite materials mimicking sound tissues.

Response

We agree entirely that the algorithm will need to be further developed to detect non-metallic (composite resin) restorations before it can be widely used to facilitate personal identification in the event of a large-scale disaster. The use of dental evidence for the personal identification of disaster victims requires that the collected dental findings are highly accurate. Since teeth with composite resin restorations can sometimes be mistaken for non-restored teeth even when observed with the naked eye, the present study limited the classification of molar tooth condition to only three categories (i.e., FMC, In and CNMR) that exhibit obvious differences in color. The accuracy of the algorithm was around 95% when these three categories were used and reached about 98% when the classification was based only on the presence or absence of a metal-colored restoration. We view our research as a first step toward the creation of a more complex algorithm that can accurately detect a wide range of tooth features, including both metallic and non-metallic restorations. Furthermore, we envisage that such an algorithm would allow postmortem dental charts to be automatically and rapidly generated from occlusal views obtained by intraoral scanning. We have completely rewritten the entire Discussion section of the manuscript, which now includes the above information (lines 243–257).

Comment

What did the authors mean by "treatment scars"?

Response

Thank you for this query. This term was intended to describe evidence of prior treatment. However, this term has been deleted from the manuscript following major revisions to the Discussion section.

Comment

The conclusions in the present form are rather an extended summary of the results, therefore should be rephrased so that they are related to the aim of the study.

Response

We have rewritten the conclusion so that it is better related to the aim of the study (lines 258–264).

---

## [Decision Letter · Decision Letter 1]

7 Dec 2021

PONE-D-21-30844R1Development of an artificial intelligence-based algorithm to classify images acquired with an intraoral scanner of individual molar teeth into three categoriesPLOS ONE

Dear Dr. ETO,

Thank you for submitting your manuscript to PLOS ONE. After careful consideration, we feel that you paper can be made suitable for publication after minor revisions. Specifically, we only ask you to address two very minor points raised by Reviewer #1 (please find them below) before proceeding to the final publication steps.

We look forward to receiving your revised manuscript.

Kind regards,

Francesco Bianconi, Ph.D.

Academic Editor

PLOS ONE

Journal Requirements:

Reviewers' comments:

Reviewer's Responses to Questions

**Comments to the Author**

1. If the authors have adequately addressed your comments raised in a previous round of review and you feel that this manuscript is now acceptable for publication, you may indicate that here to bypass the “Comments to the Author” section, enter your conflict of interest statement in the “Confidential to Editor” section, and submit your "Accept" recommendation.

Reviewer #1: All comments have been addressed

2. Is the manuscript technically sound, and do the data support the conclusions?

Reviewer #1: Yes

3. Has the statistical analysis been performed appropriately and rigorously? 

Reviewer #1: Yes

4. Have the authors made all data underlying the findings in their manuscript fully available?

Reviewer #1: Yes

5. Is the manuscript presented in an intelligible fashion and written in standard English?

Reviewer #1: Yes

6. Review Comments to the Author

Reviewer #1: The authors have addressed all the comments and now the paper is more detailed and sounds clearer for the reader.

Only two minor suggestions:

Figure 4: ‘over’ should be ‘cover’?

Line 264: it should be ‘the construction of a system’

7. PLOS authors have the option to publish the peer review history of their article (what does this mean?). If published, this will include your full peer review and any attached files.

Reviewer #1: No

---

## [Author Response · Author response to Decision Letter 1]

8 Dec 2021

We thank the reviewer for his/her careful reading of our manuscript and useful comments. Our responses to the reviewers’ comments are presented below. All revisions to the manuscript are highlighted in red-colored font and referred to by line number in the responses below. Please note that the changes made do not majorly affect the content, conclusions or framework of the paper.

REVIEWER #1

Comment

The authors have addressed all the comments and now the paper is more detailed and sounds clearer for the reader.

Response

We thank Reviewer#1 for the positive comments regarding the revised manuscript.

Comment

Figure 4: ‘over’ should be ‘cover’?

Response

Thank you for pointing out this inadvertent error. We have revised ‘over’ to ‘cover’ (Figure 4). 

Comment

Line 264: it should be ‘the construction of a system’

Response

Thank you for this helpful suggestion. We have modified the sentence ‘the construction a system’ to ‘the construction of a system’ (Line 264).

---

## [Editor Report · Decision Letter 2]

13 Dec 2021

Development of an artificial intelligence-based algorithm to classify images acquired with an intraoral scanner of individual molar teeth into three categories

PONE-D-21-30844R2

Dear Dr. ETO,

We’re pleased to inform you that your manuscript has been judged scientifically suitable for publication and will be formally accepted for publication once it meets all outstanding technical requirements.

Kind regards,

Francesco Bianconi, Ph.D.

Academic Editor

PLOS ONE
---

## [Editor Report · Acceptance letter]

30 Dec 2021

PONE-D-21-30844R2 

Development of an artificial intelligence-based algorithm to classify images acquired with an intraoral scanner of individual molar teeth into three categories 

Dear Dr. Eto:

I'm pleased to inform you that your manuscript has been deemed suitable for publication in PLOS ONE. Congratulations! Your manuscript is now with our production department. 

Kind regards, 

on behalf of

Prof. Francesco Bianconi 

Academic Editor

PLOS ONE